

# Shaping shallow landslide susceptibility as a function of rainfall events

Micol Fumagalli[1], Alberto Previati[1], Paolo Frattini[1], Giovanni B. Crosta[1]

Department of Earth and Environmental Sciences, University of Milano-Bicocca, Piazza della Scienza 4, 20126 Milano, Italy

*Correspondence to*: Micol Fumagalli (m.fumagalli86@campus.unimib.it)

**Abstract.** This paper tests a multivariate statistical model to simulate rainfall dependent susceptibility scenarios of shallow landslides. To this end, extreme rainfall events spanning from 1977 to 2021 in the Orba basin (a study area of 505 km$^2$ located in Piedmont, northern Italy), have been considered. First of all, the role of conditioning and triggering factors on the spatial pattern of shallow landslides in areas with complex geological conditions is analysed by comparing their spatial distribution and their influence within logistic regression models, with results showing that rainfall and specific lithological and geomorphological conditions exert the strongest control on the spatial pattern of landslide.

Different rainfall-based scenarios were then modelled using logistic regression models trained on different combinations of past events and evaluated using an ensemble of performance metrics. Models calibrated on multi-events outperform the ones based on a single event, since they are capable of compensating for local misleading effects that can arise from the use of a single rainfall event. The best performing developed model considers all the landslide triggering rainfall scenarios and two non-triggering intense rainfall events, with a score of 0.90 out of 1 on the multi-criteria TOPSIS-based performance index.

Finally, a new approach based on misclassification costs is proposed to account for false negatives and false positives in the predicted susceptibility maps.

Overall, this approach based on a multi-event calibration and on a misclassification costs analysis shows promise in producing rainfall dependent shallow landslide susceptibility scenarios that could be used for hazard analyses, early warning systems and to assist decision-makers in developing risk mitigation strategies.

## 1 Introduction

Shallow landslides are a widespread phenomenon that affects many regions of the world (Petley, 2012). In Italy, according to the last national report on landslides and floods, almost 8% of the country is affected by landslides, of which 15% are classified as rapid flow and 6% as shallow landslides (ISPRA, 2021). According to Cruden and Varnes (1996), these are shallow slides, mainly translational, with a thickness ranging between 0.5 and 2 m (Bandis et al., 1996; Mason and Rosenbaum, 2002). Shallow landslides are generally triggered by rainfall events, which cause an increase in pore water pressure, or a loss of apparent cohesion generated by suction (Caine, 1980; Crosta and Frattini, 2003; Fredlund et al., 1978; Iverson, 2000; Lu and Godt,



2008). Despite their limited initial volume, these landslides may be characterized by a high density per unit area and can evolve
in debris flows. The high velocity and the difficulty of prediction due to the almost complete lack of premonitory signs
(Campbell, 1975; Frattini et al., 2009; Montrasio et al., 2016) make these phenomena seriously dangerous in terms of life and
economic losses (Trigila and Iadanza, 2012).
A common definition of landslide hazard is "the probability of occurrence within a specific period of time and within a given
area of potentially damaging phenomena" (Varnes, 1984), requiring the quantification of the magnitude, the spatial and the
temporal probability for an instability event to occur. The variables that control landslide hazard are commonly distinguished
into conditioning and triggering factors. Conditioning factors are generally assumed to have no temporal dependence and are
responsible for "where" a landslide might occur, while triggering factors are event-related and control "when" a landslide might
occur (Crosta and Frattini, 2003; Lombardo et al., 2020; Wu and Sidle, 1995), although their spatial properties (e.g. distribution
of intensity or cumulative rainfall during a rain event) play a key role in determining the location of landslides.
The spatial likelihood of shallow landslide occurrence is addressed through landslide susceptibility models, based on either
physically based or machine-learning techniques. Physically based techniques for shallow landslides often combine the
infinite-slope model with hydrogeological models, which require many different input data; for this reason, they are more
frequently applied at the site-scale (Baum et al., 2008; Montgomery and Dietrich, 1994).
Machine-learning methods search for functional relationships between the conditioning factors and the distribution of
landslides, obtained from inventories of past events (Carrara, 1983; Goetz et al., 2015; Huang et al., 2020; Reichenbach et al.,
2018; van Westen et al., 2008). Susceptibility models are usually considered as time-independent, meaning that the likelihood
of landslides occurrence does not vary in time (Jones et al., 2021; Lombardo et al., 2020). However, many authors demonstrated
that this assumption is often violated both on a long (hundreds or thousands of years) and on a short timescale (tens of years),
especially in view of climate changes (Hungr, 2016; Samia et al., 2018). The "when" problem has typically been addressed by
using rainfall thresholds or Physically based models. Rainfall thresholds describe the rainfall intensity, duration or cumulative
event precipitation that may trigger landslides for a particular area (Caine, 1980; Crosta, 1998; Guzzetti et al., 2007). This
approach has usually disregarded soil features and morphometric conditioning factors, such as the geotechnical features of the
involved materials, until recent times, when hydrogeological effects started to be included into the analyses, for example
through the consideration of the soil water content prior to the triggering event (Bogaard and Greco, 2018; Marino et al.,
2020a). Some authors started testing approaches to address both the "where" and the "when" questions in the context of early
warning systems. For example, Kirschbaum and Stanley (2018), used a fuzzy overlay model to combine static explanatory
variables into a susceptibility map. This information was then incorporated into a heuristic decision tree model together with
dynamic variables such as antecedent precipitation, giving a model capable of indicating potential landslide activity in near
real-time. Segoni *et al.* (2018b), combined rainfall thresholds and susceptibility maps into a hazard matrix, while Bordoni *et*
*al.* (2021), integrated rainfall thresholds and antecedent soil humidity with a susceptibility model in order to forecast the spatial
and temporal probability occurrence of shallow landslides. Camera *et al.* (2021) included intense rainfall and snowmelt in a
landslide susceptibility model trained over multiple landslide inventories and different meteorological conditions, making it



potentially more robust to investigate the effects of climate changes. Knevels *et al.* (2020) and Maraun *et al.* (2022), included
5 days cumulated rainfall and maximum 3 hours rainfall intensity to model landslides associated with an extreme rainfall event,
and then applied their findings to an event storyline approach to analyse the future landslide occurrence probability under
climate changes. Moreno *et al.* (2024) integrated static and time-dependent controlling factors into a generalized additive
mixed model (GAMM) model to forecast shallow landslides in space and time, showing that both short-term (2 days) and
medium-term (14 days) cumulative precipitation increases the model capabilities.
Yet, the integration of static and time-varying factors into machine-learning models still remains challenging, but it could
become a powerful instrument to better understand the connection between a variation in the time-dependent controlling factors
and landslide triggering, thus helping at improving landslide prediction in a changing climate.
An important issue for the application of susceptibility models is the evaluation of their performance. For models that predict
binary stable and unstable slopes it is necessary to choose a cut-off value below which the predicted susceptibility values are
treated as 0 and above which the values are treated as 1 (Beguería, 2006; Brenning, 2005; Frattini et al., 2010; Goetz et al.,
2015; Guzzetti et al., 1999). This results in a contingency matrix quantifying the total number of correctly and incorrectly
classified units. Form this matrix, it is possible to assess the performance by using several performance statistics, such as the
Accuracy (i.e. the ratio between the correctly classified samples and the total number of samples), the Precision (i.e., the ratio
between the true positive samples and all the positively classified samples), the True Positive Rate TPR (i.e., the ratio between
the true positive and all the positives), the False Positive Rate FPR (i.e., the ratio between the false positives and all the
negatives), the Threat score (Gilbert, 1884), the Pierce's skill score (True skill statistic; Peirce, 1884), the Heidke's skill score
(Cohen's kappa; Heidke, 1926), and the odd ratio skill score (Yule's Q; Yule, 1900).
However, the choice of the cut-off value is a complex problem, and therefore the performance is frequently evaluated by using
cutoff-independent methods, such as the Receiver Operating Characteristic (ROC) curves (Frattini et al., 2010; Hosmer and
Lemeshow, 2000; Provost and Fawcett, 2001) or the Precision-Recall (PR) curves (Davis and Goadrich, 2006; Raghavan et
al., 1989; Saito and Rehmsmeier, 2015). The ROC curve represents the FPR and TPR obtained for different cutoffs. The Area
Under the Curve (AUROC) can be used to quantify the overall quality of the model (Hanley and McNeil, 1982). However,
ROC curves can overestimate the performance of a model when the distribution of the input classes is highly skewed. For this
reason, the Precision-Recall (PR) curves have also been used (Nam et al., 2024; Yordanov and Brovelli, 2020; Zhao et al.,
2022), which plots the precision (i.e., the proportion of true positives among the positive predictions) against the TPR.
However, unlike ROC curve, the value under the PR curve is not directly interpretable for model evaluation, especially because
of a non-universal baseline performance, which depends on the class distribution, and a non-linear interpolation of precision
values. Nevertheless, PR analysis can be adapted to be used similarly to the ROC analysis by using Precision-Recall-Gain
curves (PRG), which make use of the F-Gain score, a linearized version of the $F_1$ score, to properly take baselines into account
(Flach and Kull, 2015). In landslides-related problems, the quantification of the costs linked to the use of a model is also an
important issue. Therefore, the performance of the model can be done with an approach that minimize the expected





misclassification costs, through the cost curves (Drummond and Holte, 2006; Frattini et al., 2010). Moreover, the cost curve
allows to identify the optimal cut-off to be used for the performance evaluation.
A multivariate statistical analysis for the Piedmont area of the Orba basin (northern Italy) has been developed in this paper,
considering rainfall scenarios spanning from 1977 to 2021, to investigate the correlation between landslides distribution and
the spatial pattern of conditioning and triggering factors. Different logistic regression models were trained for different
landslides and rainfall scenarios, and their performance was evaluated through an ensemble of performance metrics, leading
to an optimal choice of the best model for scenario-based problems or early warning.
This work allows to address the following research questions:
•  To what extent the pattern of shallow landslides is controlled by the characteristics of the rainfall event in areas with
complex geological conditions?
•  How can rainfall be used within a statistical model to produce instability scenarios for different rainfall events?
•  Which is the best strategy to train a statistical model based on an ensemble of rainfall events?
•  Which is the most significant classification scheme to produce a susceptibility map for early warning purposes?
The novelty of this work lies in the definition of a critical selection strategy of the optimal ensemble of rainfall events to
produce a susceptibility map that may be helpful for scenario-based problems and early warning purposes. Moreover, a new
methodology is proposed for the classification of the regression results, used for the realization of the final resulting maps.
**2 Materials and methods**
**2.1 Study area**
The Orba basin is located in the Langhe and Monferrato Hills of Piedmont Region, north-western Italy. This area has been
affected by several high-magnitude floods and severe slope instabilities during the last century, caused by intense rainfall
events (Mandarino et al., 2021). The study area has an extension of 505 km$^2$ and it is situated between 80 and 1170 m a.s.l.
The main river of the basin, the Orba River, flows northward from the Ligurian Apennines to the confluence with the Bormida
River, a right tributary of the Po River. The study area overlaps metamorphic lithotypes in the southern part – mainly
peridotites, serpentinites and serpentine-schists, meta-gabbros and meta-sediments belonging to the Voltri Massif and the
Sestri-Voltaggio Zone (Piana et al., 2017) – while in the central part of the area the sedimentary sequence of the Tertiary
Piedmont Basin (TPB) outcrops. The TPB evolved from the Late Eocene to the Late Miocene over the inner part of the Alpine
wedge (Coletti et al., 2015) and is mainly represented in the area by conglomerates, sandstones and marls. The northern sector
of the basin presents quaternary fluvial deposits belonging to the Alessandria – Tortona floodplain. The morphology of the
area is strongly controlled by the TPB sedimentary succession: where the strata are harder, the landscape presents hilly reliefs
dipping in the same direction as the underlying layers, while lowered areas modelled by fluvial erosion are present where the
lithologies are more erodible. When the dipping of the strata becomes gentler, the morphology becomes more uniform and



characterized by a dense hydrographic network. The mean annual temperature is 13° and the average annual precipitation
ranges from around 600 mm/year in the northern part to 1600 mm/year in the southern part, with autumn as the rainiest season
(Fioravanti et al., 2022; Luino, 2005). Land use is primarily forest (45%), with crops and meadows (24%) near the confluence
with the Po River.

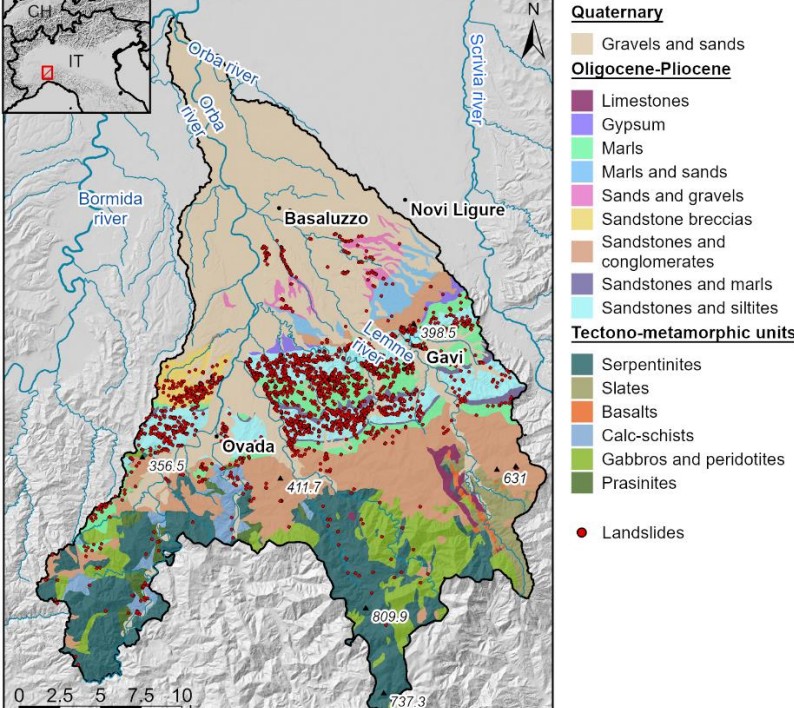


**Figure 1. Location of the Orba basin, with the spatial distribution of shallow landslide observed in three different events, and with**
**the main lithologies.**
**2.2 Data**
**2.2.1 Rainfall events and landslide inventories**
Three landslide inventories were compiled for three recent extreme rainfall events (1977, 2014 and 2019) through the analysis
of Google Earth images, national and regional orthophotos, published event maps, and field reconnaissance (Fig. 1). Part of
the inventories was already available online (SIFRAP, Sistema Informativo sulle FRane in Piemonte), while the most recent
event was provided for this project by the Regional Environmental Protection Agency of Piedmont (ARPA Piemonte, personal
communication). The 2014 and 2019 inventories include polygons of each single shallow landslide, while the 1977 inventory
represents clusters of shallow landslides as polygons. However, this difference is negligible when choosing slope units as
mapping units for the analyses (Sect. 2.3).
The first shallow landslide event was triggered by heavy rainfall at the beginning of October 1977. Between October 6th and
7th, more than 400 mm of rain fell in less than 24 hours, causing flooding, bank and riverbed erosion, debris flows and soil





slips (INTERREG IIC, 1998)(Fig. 2). The second shallow landslide event was triggered in October 2014 with more than 420
mm of rain in less than 12 hours, as recorded at the Gavi meteorological station on October 13th(Fig. 2), for which the mean
annual total rainfall is 1000 mm (calculated for the 1991 – 2020 time interval, ARPA Piemonte). The third shallow landslide
event occurred in late October 2019. In the afternoon and evening of October 21st more than 400 mm of rain (Gavi station) fell
in less than 12 hours, resulting in a very high-magnitude flood and widespread shallow landslides (ARPA Piemonte, 2019)
(Fig. 2).
In addition to these three landslide-triggering rainfall events, two intense precipitation events (2016 and 2021) that were not
associated to landslides were selected, in order to test the capabilities of the models to discriminate between triggering and
non-triggering rainfall characteristics. The 2016 event hit the Piedmont region with strong and persistent rainfalls between
November 21st and 25th, and triggered almost 1000 landslides, none of which in the Orba basin. Indeed, the peak of the
cumulative precipitation was localized more southward compared to the ones previously described, with up to 400 mm of rain
in the southern edge of the Orba basin (Fig. 2). The other event happened from October 3rd to 5th, 2021. The Ligurian-Piedmont
watershed was the most affected area, with a peak of 472 mm of rain in 12 hours recorded in the south-western part of the
area. The total precipitation in the Orba basin was up to 750 mm in the south-western edge of the basin (Fig. 2).

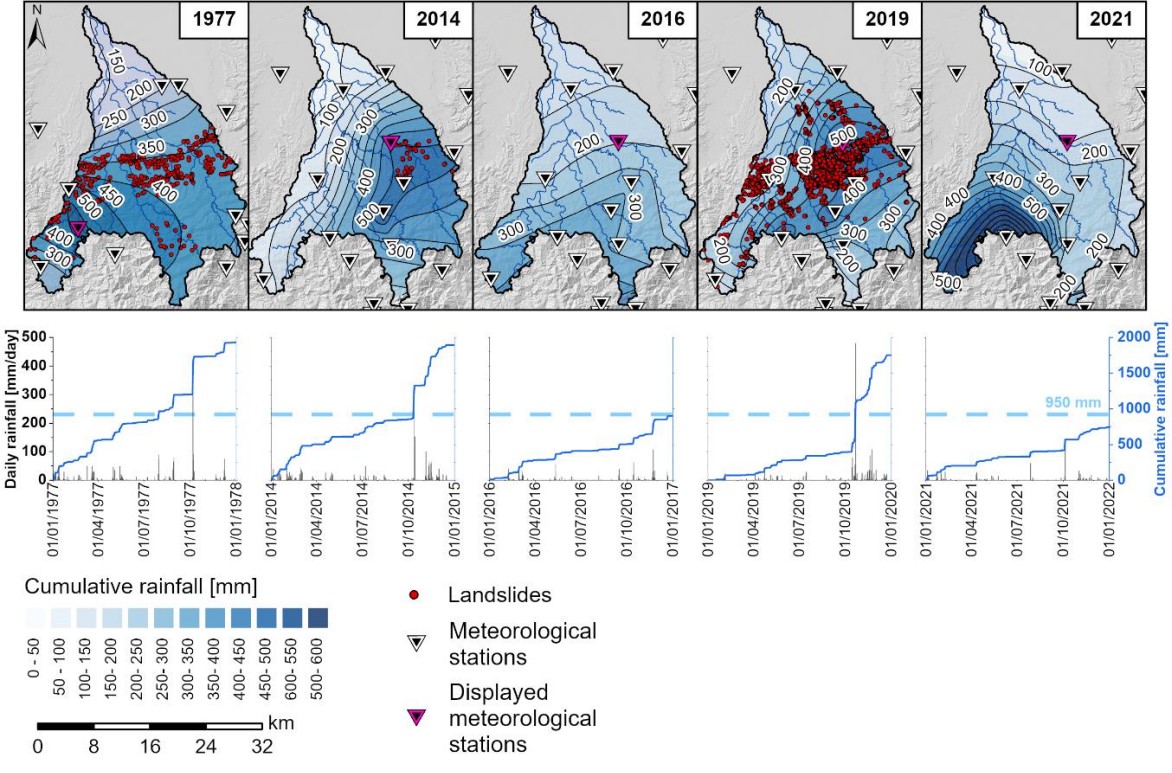


**Figure 2. Rainfall and landslides distribution during the considered events, reconstructed by interpolation of values measured by**
**the meteorological stations on the ground, that led to landslide triggering in the Orba basin. Graphs report the daily and cumulative**
**rainfall for the year in which the shallow landslides were triggered are shown. Dashed lines represent the mean annual rainfall for**
**the basin of interest (ARPA Piemonte).**




For all the inventories, a non-cumulative logarithmic binned landslide size probability density distribution was developed as:
$$p(A) = \frac{1}{N_{tot}} \frac{\partial N}{\partial A}$$ (1)
where $\partial N$ in the number of landslides with an area between A and $A + \partial A$ and $N_{tot}$ is the total number of landslides within a
study area (Malamud et al., 2004). Following (Frattini and Crosta, 2013), a Pareto distribution was fitted to the probability
density above a minimum size cut-off with (Fig. 3):
$$p(A) = \alpha c^\alpha A^{-\alpha-1} \qquad c > 0, \qquad \alpha > 0, \qquad A \, \epsilon \, [c, \infty)$$ (2)
Using the maximum likelihood estimation, the distribution parameters were estimated, obtaining a good fitting for landslides
larger than 500 m$^2$, with the best fitting results for landslides greater than 1000 m$^2$. The scaling exponents $\alpha$ vary between 1.5
and 2.6, values that are higher than most of those reported in literature but still in the range (Van Den Eeckhaut et al., 2007).

**Table 1. Statistical parameters describing the landslide events in the study area.**

| Event | Number | Density % | Total landslide area [km$^2$] | Mean landslide area [m$^2$] |
|---|---|---|---|---|
| 6 – 7 October 1977 | 366 | 1.31 | 7.82 | 21373 |
| 9 – 13 October 2014 | 66 | 0.004 | 0.023 | 353 |
| 19 – 22 October 2019 | 2088 | 0.26 | 1.57 | 124 |

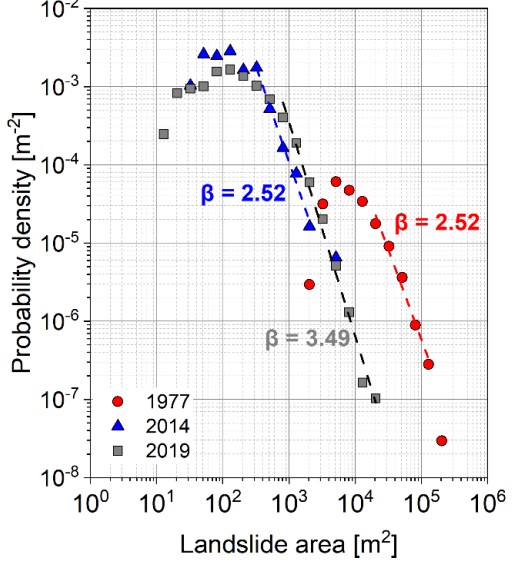

Figure 3. Probability density – areas distribution of the shallow landslides for the three events within the study area. As stated in the
main text, the 1977 landslide inventory shows a different distribution, shifted to the right, because of the different chosen mapping
criteria. Power-law fitting with maximum likely estimator is reported ($\beta$ = -$\alpha$-1).



### 2.2.2 Landslide conditioning and triggering factors

The conditioning factors used in the following analyses include 7 morphometric parameters, lithology, soil grain size distribution, and land use (Fig. S1). The morphometric parameters were extracted from a 5m resolution DEM provided by Piedmont region. The morphometric factors are slope angle, northerness, easterness, profile curvature, planar curvature, total curvature, and flow accumulation. Lithological information was obtained from the geological map of Piemonte Region, at scale 1:250,000 (Piana et al., 2017). The units have been reclassified by aggregating geo-stratigraphic units with comparable lithological and litho-technical characteristics, resulting in 16 lithological classes (Fig.1: gravels and sands, limestones, gypsum, marls, marls and sands, sands and gravels, sandstone breccias, sandstones and conglomerates, sandstones and marls, sandstones and siltites, serpentinites, slates, basalts, calcschists, gabbros and peridotites, and prasinites).

Information relative to the soils grain size distribution was retrieved from the SoilGrids maps (Poggio et al., 2021), reporting soil properties for the entire globe with a resolution of 250 m. SoilGrids models were obtained through the application of machine learning to soil data collected worldwide.

The land use was obtained from the 10 m resolution LAND COVER PIEMONTE map, which integrates information collected between 2018 and 2022 (https://geoportale.igr.piemonte.it/cms/progetti/land-cover-piemonte, last access 21/10/2023). 12 different land use classes were used, namely arable land, areas with sparse/absent vegetation, artificial non-agricultural green areas, heterogeneous agricultural areas, inland waters, mining areas, permanent crops, permanent lawns, road network, shrubby/herbaceous areas, urbanized and productive areas, and woods.

Besides the predisposing factors, several rainfall parameters potentially responsible for the shallow landslides triggering were also included into the analysis. These parameters were obtained by interpolating daily rainfall data collected at 39 and 51 gauging stations for the 1977 and 2014/2019 rainfall events, respectively. In particular, the maximum daily rainfall intensity (mm/day), the total rainfall of the events (Table 1), and the antecedent cumulative rainfall (mm) over 10, 30, 60 and 90 days (Smith et al., 2023) as a proxy of soil water content prior to the event (Guzzetti et al., 2007), which can increase the likelihood of failure (Bogaard and Greco, 2018; Thomas et al., 2018), were extracted. Maximum daily rainfall intensities were normalized by the daily rainfall with a return period of 10 years, provided by ARPA Piemonte with a grid resolution of 250 m, while the total and antecedent rainfall values were normalized by the mean annual precipitation (1991 – 2020) within the study areas. Data normalization was performed because previous studies (Marc et al., 2019; Smith et al., 2023) found that the spatial pattern of shallow landslides is more correlated with rainfall anomalies rather than with rainfall absolute values.

A correlation analysis between these rainfall variables revealed a strong linear correlation between the maximum rainfall intensity and the total rainfall of the event – probably due to the coarse temporal aggregation used to estimate the maximum intensities. A strong correlation was also found between the antecedent cumulative values over different aggregation time windows. For the subsequent regression analyses, an a priori selection was made to extract the two most influencing rainfall variables: the maximum daily rainfall intensity as an intra-event descriptor, and the 90-day cumulative rainfall for the antecedent condition. The latter was selected by testing the correlation between the cumulative rainfall values and the soil





humidity obtained from the ERA5-Land dataset (ERA5-Land hourly data from 1950 to present.; Hersbach et al., 2020; Muñoz-
Sabater et al., 2021), from which the highest correlation was found when using a time window of 90 days (Fig. S2).

**2.3 Slope unit delineation**

The application of statistical models to landslide susceptibility zoning requires the partition of the study area in terrain units,
such as unique condition units, slope units, grid-cells, or others (Carrara et al., 1991, 2008). Among these, slope units were
chosen since they provide several advantages, such as: (i) the high geomorphological meaning of the terrain unit; (ii) the
possibility to use continuous values (i.e, percentage within the unit) for the categorical variables, rather than binary values
(Carrara *et al.*, 1991), (iii) an efficient handling of possible mapping uncertainties, thanks to the generalization of the
predisposing factors falling within them (Jacobs et al., 2020; Steger et al., 2016). Their delineation is based on the identification
of drainage and divide lines, and was done automatically by using the r.slopeunits algorithm (Alvioli et al., 2016). This iterative
algorithm requires as input data the minimum circular variance for each unit, representing the allowed variability of orientation
for each grid cell belonging to the same unit, and the minimum area for each slope unit.

**2.4 Preliminary exploratory statistical analysis**

To understand which variables exert the strongest control on the landslide distribution, and if this control remains constant
through time, the distributions of the mean values of each covariate for the slope units affected by shallow landslides were
compared with the same distributions for the whole study area, and for the other inventories. The similarity among the
inventories for each covariate (i.e., the null hypothesis) is rejected if the p-value of the Dunn's test is smaller than 0.05.
To further investigate the role of antecedent and triggering precipitation, the relationship between landslide density (i.e., total
landslide area over the total slope units area) and precipitation classes (i.e., normalized maximum rainfall intensity, normalized
cumulative rainfall, and normalized antecedent cumulative rainfall) was analysed through the Spearman's rank order
correlation coefficient. Given the strong lithological control, the analysis was conducted for the entire study area and separately
for the most unstable lithological units (marls – around 30% of the total landslides number of each event, sandstones and
siltites – almost 50% of landslide in each event, sandstone breccias – 7% of landslides in 1977 and 2019, 0% in 2014, and
sandstones and marls – 4% in 1977 and 2019, 14% in 2014).

**2.5 Rainfall-based susceptibility analysis**

Binary logistic regression was chosen for the susceptibility analysis because of its widespread and validated use and because
it provides the importance of each conditioning variable in terms of standardized regression coefficients in a straightforward
manner (Carrara, 1983; Micheletti et al., 2015; Reichenbach et al., 2018).
Logistic regression describes the relationship between a binary outcome (stable or unstable unit) and a set of independent
variables (Hosmer and Lemeshow, 2000). The probability p of a sample to belong to a certain group is given by:



$ln \frac{p}{1-p} = B_0 + B_1 X_1 + B_2 X_2 + B_3 X_3 + \cdots + B_m X_m$ (3)
where Bi are the logistic coefficients, estimated from the data, that quantify the contribution of each variable Xi to the final
outcome. Logistic regression assumes that a linear relationship exists between the logit transformation of the binary outcome
and each variable selected by the model through a forward stepwise method, with a variable being included into the model if
the probability of its score statistics is smaller than an entry value of 0.05, and being removed if the probability is greater than
a removal value of 0.10. Before running the models, variables showing a strongly skewed distribution were normalized using
a log-transformation (Carrara et al., 2008), and all the static variables were then standardized using a z-score normalization
(mean equal to 0 and standard deviation equal to 1), in order to make their estimated regression coefficients comparable
(Lombardo and Mai, 2018).
Five susceptibility models were developed. Models m77, m14 and m19 were trained on a single landslide event (i.e., 1977,
2014, and 2019, respectively). The model m771419 was trained by merging all the landslide events, and finally the model
m7714161921 was trained by merging different rainfall events with or without landslides. Each dataset was divided into
training (3/4) and validation (1/4) subsets, the former being used to build the models and the latter to evaluate their predictive
performance. Each model was evaluated against itself and against all the other landslide events by using cross-validation.
Model evaluation was performed with the following strategy. First of all, two common cut-off independent methods were
applied (ROC and Precision Recall Gained (PRG) curves) to obtain their Area Under Curves. Then, the optimal cut-off
obtained by the ROC analysis was used to derive the optimal contingency matrix, from which the accuracy, precision, TPR
and FPR were calculated. Finally, these indices were summed up with a multiple attribute decision making procedure,
performed with the technique for order preference by similarity to ideal solution (TOPSIS, Hwang and Yoon, 2012), to
individuate the best model. For each model, 50 logistic regression analyses were run, in order to statistically analyse the
distribution of the susceptibility values, the regression coefficients, and the performance metrics.
To avoid an over-abundance of obviously stable units (e.g., flat areas), which would give a biased estimate of the performance,
only nontrivial units with slopes more compatible with shallow landslides triggering (>20° and < than 40°) were selected.
The economic consequences are one of the main issues in early warning; these economic costs can be significantly different
in case of false or missing alarms. This problem is usually not considered in susceptibility studies, where the classification of
susceptibility into classes (e.g. very low, low, medium, high and very high) is based on some arbitrary choice of the modeler
(Cantarino et al., 2019).
For this reason, a new practical approach to classify the susceptibility values was defined, based on the cost-curves approach.
Similarly to other methods, such as Natural Breaks (Jenks, 1967), this procedure takes into account the underlying data, instead
of using standard classes, with the advantage that it can be calibrated on a specific cost analysis.
Specifically, the cut-off corresponding to the minimum normalized expected cost was used as the centre of the third class
(medium susceptibility), and defined in this work as *half-susceptibility threshold (HST)*. The classes limits are defined based
on a geometric progression from 0 to 1, centred on HST.





Since the misclassification costs can vary significantly within the study area, and their quantification require extremely detailed
analyses, in the current work the a priori probabilities of having and not having landslides were kept equal, while three
scenarios of relative costs (Scenario 1: $c(-|+): c(+|-) = 0.5 : 0.5$, Scenario 2: $c(-|+): c(+|-) = 0.8 : 0.2$, Scenario 3:
$c(-|+): c(+|-) = 0.2 : 0.8$, where $c(-|+)$ is the cost of false negatives and $c(+|-)$ is the cost of false positives) were
considered.

## 3 Results

### 3.1 Slope units delineation

By using a minimum area of 20,000 m² and a maximum circular variance of 0.1, the study area was partitioned in 10'528 slope
units, with an average area of 56'555 m² and a maximum area of 1'868'299 m². Slope units were classified as unstable if
occupied by at least one landslide. This resulted in 627 (5.95%), 50 (0.47%), and 869 (8.25%) unstable slope units for the
1977, 2014, and 2019 events, respectively.

### 3.2 Preliminary exploratory statistical analysis

Figure 4 represents the percentage of variables within the different groups of controlling factors for which the similarity
hypothesis between the variable distributions in the unstable slope units for the different inventories can be rejected (see Fig.
S3 for all the distributions). Lithological variables show the lowest dissimilarity between the different inventories, followed
by land use. On the other side, the rainfall variables are always dissimilar among the inventories. This suggests that landslides
may be triggered by different rainfall patterns, but within certain specific lithological and land use classes.

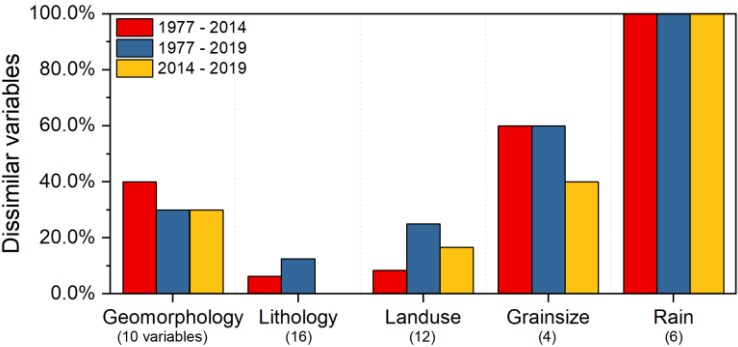


**Figure 4. Percentage of statistically dissimilar variables within each group of controlling factors, according to the Dunn's test with a significance level of 0.05.**

To further investigate the control of rainfall on landslide triggering, landslide density was plotted against classes of maximum
rainfall intensity, cumulative rainfall during the event and 90 days antecedent cumulative rainfall (Fig. 5).





Firstly, the three landslide events show significant differences, confirming the previous results. Considering the whole study
area, landslide density is clearly positively correlated with maximum rainfall intensity during the event. Interestingly, for the
same maximum rainfall intensity (Fig. 5a), the landslide density is offset for the three inventories, indicating a different
sensitivity of landslides to rainfall. This could be explained by the different levels of antecedent rainfall (Fig. 5b): the higher
the antecedent cumulative rainfall, the higher the sensitivity.
The same analysis for individual lithologies did not show clear evidences, probably due to the smaller sample of landslides in
each class (Fig. S4). This is more evident for sandstones and breccias, as this lithology is restricted to a relatively small sector
in the western part of the study area.

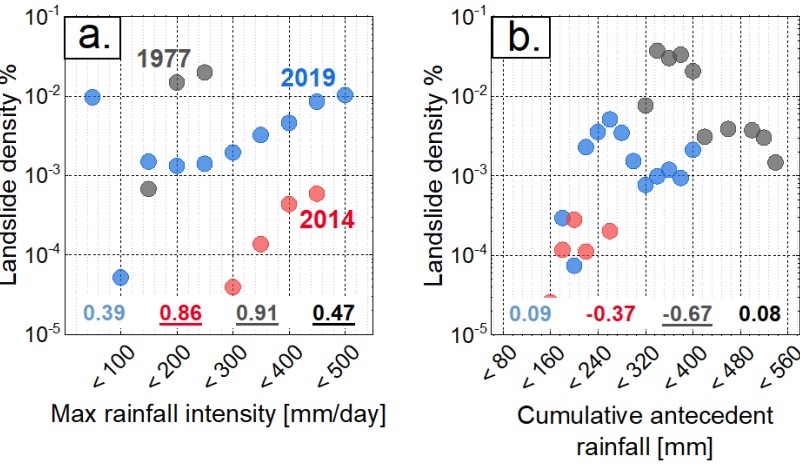

**Figure 5. Scatterplots representing landslide density in each rainfall class for the entire study area. Spearman's rank order**
**correlation coefficients between landslide density and rainfall classes are reported in each plot. Underlined values are statistically**
**correlated at the 0.05 level.**
For the 1977 and 2019 events, Fig. 5a shows that landslides started to occur for maximum rainfall intensities greater than 100
mm in 24 h, which agrees with the ID curves proposed for the area (Tiranti et al., 2019). On the other hand, during the 2014
event, a rainfall intensity of 250 mm in 24 hours was necessary to cause instabilities. This may be explained by looking at the
cumulative antecedent rainfall in 90 days, which is below 300 mm in 2014, and much higher for the other events, giving
different initial soil moisture conditions.
Also, all the three inventories show a positive correlation between the landslide density and the normalized maximum rainfall
intensity over 24 hours. On the contrary, the values of antecedent and intra-event cumulative rainfall are significantly different
between the three events (Fig. 4), as confirmed by Fig. 5. Moreover, the different average levels of antecedent conditions,
whose pattern is not spatially correlated with the distribution of maximum intensity, also play a role in offsetting the
relationship between landslide density and the maximum rainfall intensity. Figure 5 shows that landslide density increases
more rapidly with rainfall as a function of the initial conditions (for example, landslide density for 400 mm is $4.36e^{-4}$ for the
2014 event, and $4.65e^{-3}$ for 2019).



### 3.3 Rainfall based susceptibility maps

Figure 6 shows the mean coefficient and the inclusion rate of the 50 runs of the logistic regression models, for each single variable. Slope gradient is the most important parameter for all models (except m14), with always positive coefficients and a high inclusion rate. For the other morphometric parameters, northerness and flow accumulation show a high inclusion rate and relatively high coefficients (except for m14). The negative sign of the northerness coefficient indicates the south-facing slope units as more unstable. Among the lithological descriptors, "gravels and sands", "sandstones and siltites", and "marls" show the highest inclusion rates and coefficient values. On the other end, basalts, limestones, and slates are never included in the models. Land use does not exert an important control. Among the descriptors of soil granulometry, the contents in coarse fragments and sand are selected with a high inclusion rate and a negative median coefficient, with the exception of m14, while clay content is chosen with a high inclusion rate and a positive median coefficient.

Eventually, rainfall variables play an important but complex role on susceptibility. Maximum daily intensity is very important for m14, m771419, and m7714161921, with positive coefficients and a high inclusion rate. Surprisingly, maximum rainfall intensity is not included in m19, and takes negative values in m77. The antecedent cumulative rainfall is important for slope instability in models m77, m14, m771419 and m7714161921, while model m19 shows the lowest mean coefficient for this variable.

The intra event maximum rainfall intensity is also a relevant variable, but with a more complex influence. This variable is very important for model m14, with a strong destabilizing effect, but it is not included into model m19, and assumes a negative coefficient in m77.





**Figure 6. Variation of the median coefficient (left panel) and inclusion rate (frequency – right panel) of variable selection according to the different training model, based on 50 iterations. Variables are aggregated in 5 groups (G = geomorphological parameters, L = lithological parameters, S = soil grain size, U = land use and land cover parameters, R = rainfall parameters). Grey boxes indicate that the variable was never chosen by the model.**



Model m14 shows a good performance when evaluated over its validation dataset, with a mean AUROC value of 0.97 (highest
mean AUROC value among all the tested models), but it fails in predicting or hindcasting other landslide events, as indicated
by an interquartile range of AUROC values between 0.62 and 0.74 (Fig. S5), a low accuracy and a high FPR. Model m77
shows a high mean AUROC, but a low AUPRG, especially when trying to predict 2014 landslides, meaning that the model
output becomes less precise when ignoring the true negatives. On average, model m19 shows good prediction capabilities,
especially in terms of AUPRG. Models trained over multiple events show the best performance, and an associated reduction
in the variability of the final results. The mean AUROC value increases, as does the mean AUPRG. The inclusion of intense
rainfall events that did not lead to the triggering of slope instabilities results in small improvements in the general performance,
especially for the mean accuracy and FPR.
According to the TOPSIS classifier (Fig. 7), m7714161921 is the model with the highest relative closeness degree to the ideal
solution, obtained giving the same weight for the evaluation of all the scores (0.16 for all the metrics).

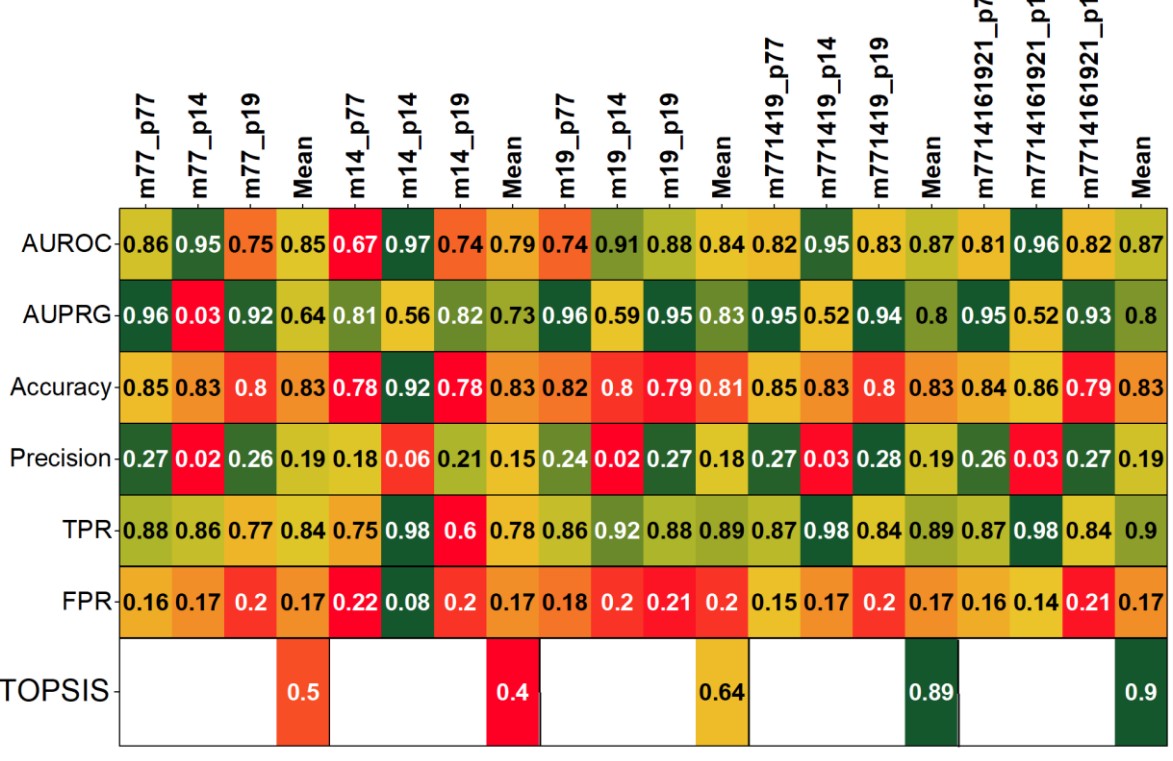


**Figure 7. AUROC, accuracy, precision, true positive rate TPR and false positive rate FPR obtained using the threshold that**
**minimizes the expected costs, calculated for each model assuming equal costs. For each model, the relative closeness degree of**
**alternatives to the ideal solution (Ci) and ranks of the evaluated models are also reported.**



## 3.4 Model representation



**Figure 8. Landslide susceptibility maps for the Orba basin. Columns refer to different training models, while rows refer to different predicted or hindcasted events.**





For each model, five rainfall events were used to produce the rainfall-based susceptibility maps (Fig. 8), obtaining different
maps for each model as a function of the event-specific rainfall values. From a simple visual inspection, comparing
susceptibility classes and landslide distribution, it is clear that models m14 and m19 are not able to correctly model landslide
susceptibility. As already seen in Fig. 6, the high coefficient of rainfall intensity in m14 makes susceptibility excessively
dependent on this variable, so that the resulting unstable units simply reflect its distribution. On the contrary, the exclusion of
rainfall intensity and the low coefficients of antecedent rainfall in m19 make the susceptibility maps almost constant for
different events. In addition, the model tends to overestimate unstable areas. Model m77 shows a better performance, but still
suffers from the low coefficient of maximum rainfall intensity, making also this model quite constant between different events,
thus predicting unstable areas also for the 2016 and 2021 events. Models m771419 and m7714161921 significantly outperform
the others, as they are able to classify the central part of the study area as unstable only for heavy rainfall events. However,
they tend to underestimate the percentage of unstable or very unstable slope units during the 1977, 2014 and 2019 events, with
less than 4% of the slope units classified as moderately, highly or very highly unstable. On the other hand, they correctly
classify all the slope units as stable when considering rainfall events that were not associated with landslides (p16 and p21).
Model m7714161921 also shows a slightly better ability to handle false positives when simulating non-triggering rainfall
events, as it can be seen in the last row of Fig. 6 for the prediction of m14, m16 and m21, especially in the western part of the
study area.
In general, the maps in Fig. 8 classified by using a rather standard partitioning of the susceptibility values into five classes (0
– 0.2, 0.2 – 0.45, 0.45 – 0.55, 0.55 – 0.8, 0.8 – 1) show an uneven distribution of slope units in the different classes, giving the
impression of either overestimation or underestimation. This problem was addressed with the new classification method based
on misclassification costs, which was applied to m7714161921 (ranked as the best performing model). For each of the three
considered scenarios the optimal cut-off threshold and the relative geometric progression were derived, considering different
misclassification cost ratios (Table 2). The class boundaries derived from the geometric progression were then used to
reclassify the susceptibility values, to produce optimised maps (Fig. 9). The optimal cut-off threshold decreases as the relative
cost of false negatives decreases, thus reducing the number of slope units classified as unstable.
**Table 2. Threshold values for m7714161921, for each of the proposed scenarios of relative costs. HST is the half-susceptibility**
**threshold corresponding to the value that minimizes the normalized expected cost for each cost scenario.**

| Cost Scenarios | HST | VL | L | M | H | VH |
|---|---|---|---|---|---|---|
| $c(-\|+): c(+\|-) = \mathbf{0.5 : 0.5}$ | 0.034 | 0.005 | 0.018 | 0.068 | 0.261 | 1.000 |
| $c(-\|+): c(+\|-) = \mathbf{0.8 : 0.2}$ | 0.010 | 0.018 | 0.068 | 0.261 | 1.000 | 0.005 |
| $c(-\|+): c(+\|-) = \mathbf{0.2 : 0.8}$ | 0.104 | 0.066 | 0.164 | 0.405 | 1.000 | 0.027 |


**Figure 9. Instability maps relative to the best performing model (m7714161921). Each row refers to a different relative cost scenario, where the proportions refer to the ratio between costs associated to false negatives and false positives. Classes limits are defined based on the optimal cut-off threshold and the relative geometric progression.**





## 4 Discussion

### 4.1 Landslide distribution analysis and prediction

This paper investigated the relationship between several spatially distributed variables (i.e., possible triggering factors) and the occurrence of shallow landslides though a logistic regression-based susceptibility analysis.

At a first visual inspection, the spatial distribution of the shallow landslides is fairly constant in all the available inventories, suggesting that shallow landslides in this area are modulated by rainfall, but controlled by other static parameters. In particular, landslides tend to occur in slope units with similar geomorphology and lithology (Fig. 4).

More specifically, beside the slope gradient, lithology appears to be the most important variable that controls landslide susceptibility (Fig. 6). Among all, the most prone lithologies in all the inventories are marls, sandstones and siltites, similarly to the results of Luino and Padano (1999) and Licata *et al.* (2023). The high importance given to gravels and sands, a lithology commonly found in alluvial flat areas, can be explained with the instability of fluvial terraces (Šilhán, 2022). Moreover, the lithology controls the grain size of the soil cover and, thus, the hydrological processes in the unsaturated zone. The sedimentary sequence in the central part of the area, overlaid by soils with a high clay content, is another important destabilizing factor in the model because of the poor draining capacity of clays. More interestingly, the southern metamorphic basement is commonly covered by soils rich in sand and coarse fragments, which have a strong stabilizing effect, probably correlated with a higher drainage capacity and friction angle.

Surprisingly, the role of land use does not appear to be relevant. In addition, the role of lithology may be strong enough to mask the land-use effect.

Looking at the variables related to rainfall dynamics, the cumulative antecedent rainfall is the most relevant in all regression models. In fact, it has been considered a proxy for the soil water content before the event, which for various authors it is pivotal for modelling shallow landslides (Bogaard and Greco, 2018; Marino et al., 2020b). The intra event maximum rainfall intensity is also a relevant variable, but with a more complex influence. Being calculated with a 24 h aggregation time, this variable can be intended as a general descriptor of the entire rainfall event, representative of both the rainfall intensity and the daily cumulative value. Using a smaller aggregation time could help to differentiate the effects of these two descriptors, which was impossible for the event of 1977, as outlined in Sect. 4.2.

These parameters are also important to explain the spatial distribution of the landslide density. In particular, the analysis of the relationships between landslide density, the normalized maximum rainfall intensity over 24 hours and the normalized values of the antecedent cumulative rainfall suggest that landslide density appears to be controlled by the maximum rainfall intensity. This agrees with the mechanical explanation of shallow landslides triggering, controlled by soil saturation, leading to an increase in pore pressure and a loss of soil suction (Fredlund et al., 1978). In addition, the antecedent condition shows a double role of setting a threshold required for landslide initiation (e.g. Crozier, 1999; Glade *et al.*, 2000; Godt *et al.*, 2006; Marino *et al.*, 2020b), and offsetting the relationship between landslide density and rainfall intensity.





Several basin-scale studies suggest that to quantify the shallow landslide susceptibility, the use of multitemporal inventories
lead to better results (Reichenbach et al., 2018), while others affirm that this is not always associated with a model performance
improvement (Ozturk et al., 2021; Smith et al., 2021). Results show that, for the Orba basin, models trained over a single
landslide event are not capable of catching the real processes underlying the instability phenomena, despite the high landslide
density and the good performance when using the test dataset. Thus, they are unable to predict landslide events associated with
different rainfall characteristics. In particular, m14, being the smaller landslide inventory and more limited in the extend of the
affected area, shows the best performance when tested against itself (Fig. 7 and Fig. S5), and the worst performance when used
to model other events, producing maps with an exaggerated landslide susceptibility in areas with high precipitation. The
inclusion of multiple events helps in stabilising the effect exerted by the different controlling variables, thus providing more
reliable prediction/hindcast susceptibility maps.
The evaluation of the performance of regression models is always challenging, especially when using an input dataset with a
skewed distribution (e.g. Provost *et al.*, 1998; Davis and Goadrich, 2006; Drummond and Holte, 2006). AUROC, which is the
most used evaluation method in the literature (Reichenbach et al., 2018) suffers from an overly optimistic evaluation while
misclassifying the samples that belongs to the underrepresented class. This is the case of model m77 when predicting 2014
event. On the other side, AUPRG shows high values when model m77 predicts 2019 event, even if large parts of the area
affected by landslides is predicted as stable. The other indices are cut-off dependent, and they do not show any capabilities to
discriminate among the different models. For these reasons, the multi-criteria TOPSIS model was used to consider the
contribution of all the indices. Interestingly, the TOPSIS classification shows significant variations across the models where
single appear to show no significance. Based on the TOPSIS evaluation, the multitemporal models outperform the single event
models, confirming what discussed above. In particular, the model with the highest prediction capabilities is m7714161921,
suggesting that the inclusion of non-triggering rainfall events helps in defining the rainfall threshold to trigger instabilities in
different parts of the study area.
For the representation of the results, the classification scheme typically adopted in the literature does not account for
misclassification costs (Cantarino et al., 2019), which are implicitly assumed equal. However, since the misclassifications
costs are often not equal, the total misclassification cost can be reduced by playing on the degree of conservativeness of the
models in order to reduce the false negatives or false positives rates, thus increasing or decreasing what is classified as unstable.
This required a new classification scheme to adjust the thresholds used for susceptibility classification according to the selected
proportion of misclassification costs.
Scenario 2, where the costs of false negatives are higher, is the most conservative because the classification is forced towards
instability to keep the false negatives rate low. On the contrary, scenario 3, where the costs of false positives are higher, shows
the highest percentage of stable slope units. Scenario 1 considers equal costs for false positives and false negatives, and
produces intermediate results. The strong differences between these scenarios suggest that the use of cost curves for the
landslide susceptibility model could be a valuable tool in the final stages of a susceptibility analysis, when slope units need to
be classified. This approach allows for different classification thresholds based on cost combinations, enabling the evaluation





of their consequences. Costs may include direct costs like damage to infrastructure and loss of life, and indirect costs like
traffic disruptions and lost productivity (Sala et al., 2021). While this work uses different cost ratio scenarios to demonstrate
the approach's potential, more detailed analyses could provide precise cost quantifications, considering that costs may vary
across different parts of the study area.

**4.2 Challenges, uncertainties, and limitations**

It is necessary to underline possible uncertainties and assumptions regarding the input datasets and the modelling strategies,
so that the limitations of our findings are made clear. Two main limitations can potentially affect the results of these analyses:
the consideration of land use and land cover as a static variable and the use of an old landslide inventory.
First, land use and land cover can vary greatly over time. Considering this variable as static is mainly due to a lack of
information, since the only other dataset provided by ARPA Piemonte dates to 2010, and the analysis of satellite images,
besides being beyond the purpose of this study, was not possible for the 1977 event. An analysis of the land use change between
two available datasets (2010 and 2021) within the Orba basin revealed that permanent crops decreased by 6% and meadows
by almost 2%, while the areas characterised by shrub and herbaceous vegetation increased by 4% and the woods by almost
4%. However, these changes can be considered negligible in the analyses, given the very low influence of the land use variables
in the logistic regression. This is in contrast with the conclusions of many other studies (for example Bernardie *et al.*, 2017;
Persichillo *et al.*, 2017; Hürlimann *et al.*, 2022), suggesting that this relationship could be further analysed in future studies.
The second limitation is posed by the inclusion of an older event (1977) with higher uncertainty of both rainfall pattern and
landslide distribution. Data from the ARPA Piemonte and ARPA Liguria weather stations were used to analyse the rainfall
pattern. However, only 36 stations were active in 1977, 26% less than in 2014 and 2019, and most of them are located outside
the region of interest (Fig. 2). This uncertainty in the rainfall pattern could affect the modelling, especially in the central part
of the basin. In addition, data for 1977 were only available with a daily time step, making it impossible to use multiple different
aggregation times. The landslide inventory of the 1977 event represents landslides as areas affected by diffuse shallow
landslides rather than individual polygons. This affects the landslide distribution and density analysis. However, the choice to
use slope units for analysis mitigated this difference in the inventories. Finally, as mentioned above, the use of this landslide
event precluded the use of satellite products; therefore, some factors that could improve susceptibility analyses, such as
satellite-based antecedent soil moisture, could not be incorporated into the model.

**5 Conclusions**

This study demonstrates the feasibility of using logistic regression to model the effects of extreme rainfall events on the stability
of a complex study area, such as the Orba basin in the Piedmont region of Italy. In this area, the spatial distribution of shallow
landslides reflects the distribution of lithology and geomorphology, thus showing a similar pattern for different rainfall
scenarios.





In such conditions, the development of a rainfall dependent model capable of simulating different susceptibility scenarios is
more challenging, and requires a careful calibration of the model with representative and significant rainfall events over a
multi-temporal dataset. In fact, the use of single events may be problematic. For example, a rainfall event that is spatially
concentrated in a small area with specific geological characteristics (such as in 2019 for the study area) could overestimate the
role of such characteristics despite the rainfall, producing biased scenarios. On the contrary, a model trained on an extreme
localised event spanning different geological conditions (such as the 2014 event) may overestimate the role of rainfall at the
expense of geology. Finally, a rainfall event evenly distributed over the area (such as in 1977) would produce a model that
underestimates the role of rainfall in controlling the landslide pattern.
To avoid such effects, an ensemble of rainfall events is preferable to better unravel the effects of the triggering variables, and
also to compensate for local misleading effects that may arise from the use of a single rainfall event. The use of rainfall events
that did not trigger landslides may also be helpful for such compensation. The proposed strategy for selecting the best ensemble
of rainfall events was based on the maximization of the AUROC, AUPRG, accuracy and precision, and the minimization of
the expected misclassification costs.
Eventually, misclassification costs were adopted as a criterion to define the susceptibility classes for the practical use of the
resulting maps; this highlights the need to give importance to the classification process, which should be tailored to the needs
of the end users and on the purpose of the final products.

**Data availability**

Data for Regione Piemonte are publicly accessible at https://geoportale.igr.piemonte.it/cms/ (Geoportale Regione Piemonte)
and at https://www.arpa.piemonte.it/ (ARPA Piemonte).

**Supplement link**

The supplement related to this article is available online.

**Author contribution**

MF: conceptualization, data preparation, analysis/coding, writing – original draft. AP: conceptualization, visualization, writing
– review and editing. PF: conceptualization, validation, writing – original draft. GC: conceptualization, writing – review and
editing.

**Competing interests**

The authors declare that they have no conflict of interest.



**Disclaimer**

Publisher's note: Copernicus Publications remains neutral with regard to jurisdictional claims in published maps and institutional affiliations.

**Acknowledgements**

The authors would like to thank Luca Lanteri and the Operative Group "Landslides monitoring and geological studies" of the Regional Environmental Protection Agency of Piedmont for the sharing of landslide mapping data related to the 2019 landslide event.



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
