# Peer review of "Shaping shallow landslide susceptibility as a function of rainfall events"

_Natural Hazards and Earth System Sciences, 2024_

## Author Comment (AC1)

**Response to Review 1 of "Shaping shallow landslide susceptibility as a function of rainfall events" by Fumagalli et al.**

We would like to thank the editor and the two reviewers for the detailed and valuable comments on this manuscript. We revised the entire draft and, based on the given suggestions, made changes within some paragraphs, to the indicated figures, and to the cross-references.

Below we address the specific comments and questions made by the 1 reviewer (for lines numbering please refer to the original manuscript).

Looking forward to your replies,

Micol Fumagalli, on behalf of all co-authors
* * *
This manuscript deals with the high frequency–low magnitude phenomena of shallow landslides that occur after heavy rainstorms at the northern edge of the Ligurian Apennines in Italy. A set of 5 events (3 with landslides, 2 without landslides) are used to identify the conditioning and triggering factors for shallow landslides with emphasis on rainfall. The authors first use multiple logistic regressions to develop five different models, whose performance they evaluate based on a method called TOPSIS that accounts for a set of classification quality criteria. The final definition of susceptibility thresholds for their best model, incorporates the economic costs of misclassification for three different scenarios. I.e. they come up with three different susceptibility maps for each of the 5 rainfall events.

I agree that their approach would be helpful in early warning and mitigation of such small but still dangerous landslides under future rainfall scenarios.

Thank you for your positive opinion on the proposed approach.

The main points I took away from this study was that (i) the best model was trained using the landslide and rainfall data from all 5 events, (ii) including heavy rainfall events that did not trigger landslides helps to account for the rainfall threshold below which landslides are less likely, (iii) the pre-event accumulated rainfall as a proxy for the soil saturation is an important conditioning factor and (iv) that the susceptibility maps are sensitive to the assumed misclassification cost scenario.

The manuscript is in fairly good shape but needs some improvement concerning the description of the data and the methodology and a harmonization between text and figures/tables. For example, I think it would have helped to include a map of the slope units used for the analysis at least in the supplements. At several places the authors refer to information in a figure/table that is actually missing. After addressing these and the following points, I think the manuscript would be ready for publication.

Jürgen Mey

**More specific comments:**

L34: "a" missing

L34 will be changed accordingly.

L78: the difference between "Precision" and TPR is not clear

L 76 – 81 will be changed into

"From this matrix, it is possible to assess the performance by using several performance statistics, such as the Accuracy (i.e. the ratio between the correctly classified samples and the total number of samples), the Precision (i.e., the ratio between the true positive samples and all the positively classified samples, meaning the sum of the true Positives and the False Positives), the True Positive Rate TPR (i.e., the ratio between the true positive and all the positives, meaning the sum of the True Positives and the False Negatives), the False Positive Rate FPR (i.e., the ratio between the false positives and all the negatives), the Threat score (Gilbert, 1884), the Pierce's skill score (True skill statistic; Peirce, 1884), the Heidke's skill score (Cohen's kappa; Heidke, 1926), and the odd ratio skill score (Yule's Q; Yule, 1900)".

Also, formulas for these indices will be added to Fig.7 (Fig.8 in the revised version).

L94: You should define here what you mean by "costs".

To better clarify the concept of false and missed alarms, and their associated costs, L94 – 95 will be extended as follows:

"One important consequence of the choice of the cut-off value is the generation of false and missed alarms, meaning the situations in which the model predicts a landslide in a specific area or time, but no landslide actually occurs, or the case in which a landslide takes place, but the model fails to predict it. False and missed alarms come with associated costs. For example, false alarms may lead to unnecessary evacuations or resources allocation, and can reduce trust in the model capabilities, while missed alarms result in unpreparedness and potentially severe consequences, including property damage, loss of life, or economic impacts."

L119: peridotite is not a metamorphic rock

L118 will be changed into

"The study area overlaps magmatic and metamorphic lithotypes in the southern part"

L125: I doubt that "dipping" is the right terminus for describing the orientation of relief. In fact, relief has no orientation at all. You write about the strata (also in the following sentence) but you do not mention the actual orientation in terms of strike and dip. It would be interesting whether (or where) the strata (or other discontinuities) are oriented parallel to the hillslopes.

The relief in the area is controlled by the different geological/geomechanical rock characteristics, and in particular by the presence of a monoclinal structure striking WNW-ESE dipping at approximately 30°.

L123-126 will be modified as follows:

"The morphology of the area is strongly controlled by the TPB sedimentary succession: where the strata are harder, the landscape presents hilly reliefs with an asymmetric profile resulting from the bedding of marly-silty and sandy-arenaceous alternations, which are part of a monoclinal structure striking WNW-ESE that imposes a dipping of approximately 30° (Luino, 1999; Mason and Rosenbaum, 2002), while lowered areas modelled by fluvial erosion are present where the lithologies are more erodible."

L129-130: give reference

This information was obtained by the authors for this manuscript through the analysis of the land use map that was provided by the regional authorities. Reference to the dataset will be added in L130.

L136: You write that you have analyzed GE images, orthophotos, event maps and field observations but your manuscript lacks any description of how you analyzed GE images, orthophotos etc.. What do you mean by field reconnaissance? Have you done field work yourselves or do you refer to the work of others?

We agree that this point was unclear into the manuscript. All the inventories were realized by Regional Environmental Protection Agency of Piemonte. The inventories related to the events of 1977 and 2014 were already available online (SIFRAP, Sistema Informativo sulle FRane in Piemonte), and a short description of how they were compiled was available. The same methodology was used for the 2019 event, for which the inventory has not been published yet. We simply visually checked for the correspondence of the mapped polygons with landslides visible on the available historical images in GE and other aerial photos.

To clarify this, we will change L 136 – 140 into:

"The inventories related to three different landslide events happened in 1977, 2014 and 2019 were used for the subsequent analyses. Data relative to the events of 1977 and 2014 are available online (SIFRAP, Sistema Informativo sulle FRane in Piemonte, handled by Regional Environmental Protection Agency of Piemonte – ARPA Piemonte) and were compiled through the analysis of Google Earth images, national and regional orthophotos, published event maps, and field reconnaissance, while the most recent event was directly provided for this project by ARPA Piemonte (personal communication)."

L141-142: Disregarding the totally unclear (for me) use of "slope units" as "mapping units", you obviously jump to the conclusion that the difference is negligible before justification is given.

L141-142 will be removed, while the description of slope units will be added, as suggested in one of the following comments.

L167: "(Frattini and Crosta, 2013)" not in parentheses

L167 will be changed accordingly.

L181: What kind of DEM is this? Please give a reference. Which software did you use to extract the morphometric parameters from the DEM?

The DEM we used is a DTM that was acquired using a uniform methodology (LIDAR) at Level 4 standard. The grid resolution (spacing) is 5 m, with an elevation accuracy of ±0.30 m (±0.60 m in areas of lower precision, corresponding to wooded and densely urbanized areas). The morphometric parameters were extracted using ArcGIS Pro 3.1.0 ©.

L 181 – 182 will be changed into:

"The morphometric parameters were extracted using ArcGIS Pro 3.1.0 © from a 5m resolution DTM acquired using a uniform methodology (LiDAR) at Level 4 standard, with an elevation accuracy of ±0.30 m (±0.60 m in areas of lower precision, corresponding to wooded and densely urbanized areas), provided by Piedmont region."

L184-187: It would be good to have a table that shows exactly, which units from the original map have been aggregated.

We report the table showing the way in which the original lithological units were aggregated at the end of the reviewing comments. The table will also be added in the supplementary material (Table S1).

L188-190: Given that this is a global database, is there any validation data (ground truth) available for the SoilGrids map in your study area?

Within the study area, there is only one soil profile in the official WoSIS Soil Profile Database, the database used for the generation of SoilGrids soil property maps, that has been used to obtain the gridded maps. Moreover, 10 soil samples were collected within the municipality of Gavi and could be used for ground truth. However, this information is very punctual and the comparison can give only a qualitative idea of the precision of the SoilGrids database.

The table below reports a comparison of the information obtained from SoilGrids and from the sample analyses, for a soil depth between 0 and 20 cm. Limits between size classes follow the MIT standard. The SoilGrids data tend to slightly overestimate the clay percentage and to underestimate the coarser fractions, while it recognized the silt class as the most abundant. However, given the different size of the reference area, the SoilGrids data were considered as reliable.

| Source | % Gravel (> 2 mm) | % Sand (>0.06 mm) | % Silt (> 0.002 and < 0.06 mm) | % Clay (<0.002) |
|---|---|---|---|---|
| SoilGrids (gridded data) | 14 | 14.62 | 45.58 | 25.8 |
| Gavi samples | 8.57 | 33.83 | 51.72 | 5.88 |

L197: How did you interpolate the rainfall data and to what resolution?

Rainfall data were interpolated using the Natural Neighbour algorithm (Sibson, 1981) at a resolution of 5 m, to match the resolution of the topographical maps. Natural Neighbour preserves the original value at the sample points and is resistant to biases that could be introduced when sample data form clusters.

L197 will be modified as follows:

"These parameters were obtained by interpolating daily rainfall data collected at 39 and 51 gauging stations for the 1977 and 2014/2019 rainfall events, respectively, with a natural neighbour technique, at a spatial resolution of 5 m."

L199: Table 1 is not showing the "total rainfall of the events"

L199 the wrong cross-reference to Table 1 will be removed. The maximum daily rainfall intensity for each event will be reported in the new Fig. 2, while the cumulative antecedent values for each rainfall event will be added into the supplementary material (Fig S2).

L201: Maximum daily rainfall intensities for each event?

L198-201 will be modified as follows:

"In particular, the maximum daily rainfall intensity (mm/day) and the antecedent cumulative rainfall (mm) over 10, 30, 60 and 90 days (Smith et al., 2023) as a proxy of soil water content prior to the event (Guzzetti et al., 2007), which can increase the likelihood of failure (Bogaard and Greco, 2018; Thomas et al., 2018), were extracted for each event."

L202: Which values did you use for the daily rainfall with a return period of 10 years and the mean annual precipitation?

Rainfall data used for normalization were obtained from the ARPA Piemonte regional analyses (ARPA Piemonte and Regione Piemonte, 2020). In particular, values for a daily rainfall with a return period of year were calculated by fitting a GEV distribution to the observed data relative to the 24h

time interval, while the mean annual precipitation values were calculated as the annual average of daily cumulative precipitation, calculate over the period 1991-2020. The spatial distribution of these normalization values will be added in the supplementary material (Fig. S3).

L203: "areas": Do you really use different study areas for the events?

No, we used only one study area. L201-203 will be modified as follows:

"Maximum daily rainfall intensities were normalized by the daily rainfall with a return period of 10 years, provided by ARPA Piemonte with a grid resolution of 250 m, while the total and antecedent rainfall values were normalized by the mean annual precipitation (1991 – 2020) within the study area."

L215-223: I am struggling with this paragraph. Can you give a definition of slope unit? Is this a subset of hillslopes that have a certain range of orientation in terms of azimuth and inclination? Since slope unit is a terrain unit I suggest rephrasing (i). What is meant by the "percentage within a unit"? Percentage of grain sizes or lithological units etc.?

To assess the susceptibility of a specific area, it is essential to divide the territory into territorial units, which are homogeneous portions of space that maximise the internal homogeneity and the external heterogeneity, to which a specific level of susceptibility can be assigned. Ideally, territorial units should be homogeneous, easily recognisable, geomorphologically significant, and objectively defined. There are different partitioning units, such as the unique condition units, the slope units and pixels. In particular, a slope unit is a morphological terrain unit delimited by drainage and divide lines (Carrara et al., 1991; Guzzetti et al., 1999), and corresponds to what could be defined as a single slope, a combination of adjacent slopes, or a small catchment, from a geomorphological and a hydrological point of view (Alvioli et al., 2016). The delineation of slope units starts with the definition of the boundaries of hydrological "half-basins", which are then grouped considering the variability of terrain aspect, meaning that half-basins with a different average terrain aspect will be considered as belonging to different slope units.

When dealing with categorical variables, such as lithological units or land use information, the advantage of using slope units instead of pixels is that, instead of using a presence/absence information, it is possible to convert the categorical values into percentages within the slope unit. For example, if ¼ of the slope unit is covered with grass, ¼ with shrubs and ½ with woods, then those categorical classes could be converted into percentages 25%, 25%, 50%. In this way, the value that one of these variables can assume is not 0/1 but can vary on a continuous scale between 0 and 100.

In light of this, paragraph 2.3 will be modified as follows:

"The application of statistical models to landslide susceptibility zoning requires the partition of the study area in terrain units, such as unique condition units, slope units, grid-cells, or others (Carrara et al., 1991, 2008). Among these, slope units were chosen for area partitioning within this study. A slope unit is defined as a morphological terrain unit delimited by drainage and divide lines (Carrara et al., 1991; Guzzetti et al., 1999), corresponding to what could be defined as a single slope, a combination of adjacent slopes, or a small catchment from a geomorphological and a hydrological point of view (Alvioli et al., 2016). Slope units were selected since they provide several advantages, such as: (i) the reproducibility of the spatial partitioning; (ii) the possibility to use continuous values for the categorical variables, where the continuous values are calculated as the areal percentage of the slope units that is covered by a particular categorical class, and thus can vary between 0% and 100% (Carrara *et al.*, 1991), (iii) an efficient handling of mapping uncertainties, thanks to the generalization of the predisposing factors falling within them (Jacobs et al., 2020; Steger et al., 2016).

Their delineation is based on the identification of drainage and divide lines, and was done automatically by using the r.slopeunits algorithm (Alvioli et al., 2016). This iterative algorithm requires as input data the minimum circular variance for each unit, representing the allowed variability of orientation for each grid cell belonging to the same unit, and the minimum area for each slope unit."

L232-235: Isn't this already part of the results?

Yes, this line reports a result and will be consequently moved within paragraph 3.2.

L259: Which indices? The latter four?

All the performance metrics, meaning the value under the ROC and PRG curves and the accuracy, precision, TPR and FPR were used for the final evaluation.

L259-261 will be modified for clarification:

"Finally, the two values under the ROC and PRG curves and the four performance metrics calculated from the contingency matrix were summed up with a multiple attribute decision making procedure, performed with the technique for order preference by similarity to ideal solution (TOPSIS, Hwang and Yoon, 2012), to individuate the best model."

L261: What is the difference between these 50 analyses? Do you change the training and validation subsets?

For each of the 50 analyses the training and validation dataset were randomly extracted from the original dataset.

L261-262 will be modified as follows:

"For each model, 50 logistic regression analyses were run with different training and validation datasets, randomly extracted from the original database. This procedure lead to the calculation of 50 different values of the coefficient associated with each controlling variable, and to the generation of 50 different susceptibility maps, thus allowing to statistically analyse the distribution of the susceptibility values, the regression coefficients, and the performance metrics."

L295-296: Fig. 5 does not show the cumulative rainfall during the event.

That part will be removed from L295-296.

L297-301: This paragraph is hard to comprehend given Fig. 5. What do the individual points in Fig.5 represent? Why are there only 3 of them for the 1977 event in panel (a) but 10 for the 2019 event.

The individual points represent the landslide density within each rainfall class, meaning that for each event we individuated the areas in which the maximum rainfall intensity or the antecedent cumulative rainfall fell in the same interval, and within those areas we calculated the landslide density (please see below). Only three points are available for the 1977 event because the local maximum daily rainfall (daily intensity) ranged between 100 and 250 mm/day, so only three classes could be obtained for this dataset (see contour in Fig.2). Conversely, for the 2019 event the range of values was much wider, and so we could calculate landslide density within more classes.

L298-299: "..for the same maximum rainfall intensity (Fig. 5a), the landslide density is offset for the three inventories.." This cannot be judged from the figure because events 1977 and 2014 have not a single max rainfall intensity data point in common.

Information about classes within which no landslides occurred was not visible into the graphs because of the logarithmic y-axis scale. The axis will be changed so that also those values will be visible, with the different inventories actually showing an overlap.

L300-301: "the higher the antecedent cumulative rainfall, the higher the sensitivity" à In Fig. 1 the cumulative rainfall curves show a very similar 90 days antecedent cumulative rainfall for 2014 and 2019. Why do these events come up so different in panel (b).

Fig.2 reported the cumulative rainfall values during a certain rainfall event, which could have a different duration. To make results more comparable, this quantity has never been used in the analyses and therefore Fig.2 will be modified by reporting the maximum rainfall intensity values for each event.

L309-310: But there is one point in Fig. 5a below 100 mm/day for the 2019 event.

What does ID stand for?

The interpretation of the plot will be slightly changed in light of this. ID stands for Intensity-Duration curves, meaning the threshold curves individuating the minimum rainfall intensity associated to a certain duration of a rainfall event that is necessary to trigger shallow landslides within a certain area.

Considering all the comments related to paragraph 3.2, it will be partially rewritten:

"To further investigate the control exerted by rainfall on the triggering of shallow landslides, the correlation between landslide distribution and values of maximum rainfall intensity and 90-days antecedent cumulative rainfall was analysed. This investigation was carried out by defining intervals of rainfall values and calculating the spatial density of landslides within each rainfall interval area. The three landslide events show significant differences, confirming the previous results. Considering the whole study area, landslide density is clearly positively correlated with maximum rainfall intensity. For the same maximum rainfall intensity values (Fig. 6a), the landslide density is offset for the three inventories, suggesting a different sensitivity of landslides to rainfall (for example, landslide density for 400 mm is $4.36e^{-4}$ for the 2014 event, and $4.65e^{-3}$ for 2019). This could be explained by the different levels of antecedent rainfall (Fig. 6b): the higher the antecedent cumulative rainfall, the higher the sensitivity. This relationship is recognizable also by visual comparison of the event rainfall intensity maps with respect to the antecedent cumulative rainfall maps (Fig. 2 and S2).

The same analysis was conducted for the most unstable lithological units, namely marls (around 30% of the total landslides number for each event), sandstones and siltstones (almost 50% of landslide in each event), sandstone breccias (7% of landslides in 1977 and 2019, 0% in 2014), and sandstones and marls (4% in 1977 and 2019, 14% in 2014). The results did not show clear trends, probably due to the small number of landslides in each rainfall class (Fig. S6). This is more evident for sandstone breccias, as this lithology is restricted to a relatively small sector in the western part of the study area.

For the 1977 event, Fig. 6a shows that landslides started to occur for maximum rainfall intensities greater than 100 mm in 24 h. This result agrees with the Intensity-Duration (ID) threshold curves proposed for the area (Tiranti et al., 2019). A few landslides in 2019 were triggered at even lower rainfall values, very close to the catchment divide where local topography could have exerted a major control. The high density is also related to the small catchment area pertaining to the low rainfall interval. On the other hand, during the 2014 event, a rainfall intensity of 250 mm in 24 hours was

necessary to cause instabilities. This may be explained by a relatively low cumulative antecedent rainfall (below 300 mm) with respect to the other events, inducing low initial soil moisture conditions."

L358: What is (Ci)? You do not use it at any other place, so I suggest to delete it. How are the ranks of the evaluated models reported? I don't see it.

Following your suggestion, Ci, representing the relative closeness degree of each alternative to the ideal solution, will be removed from the caption. The ranks (now corrected with *scores*) of the model are calculated considering the average values for each evaluator, calculated as the mean between the values obtained for each model, validated against different datasets.

L429: First use of "test dataset". Can you define this earlier?

Test dataset was erroneously used as a synonym of "validation dataset". L429 will be modified with "validation dataset".

L442: "Interesting…" I do not understand this sentence. Suggest rephrasing.

L442-443 will be removed.

L448: I guess you mean that the FNR and FPR costs are assumed to be equal.

Yes, L448 is referred to the costs associated with false positives and false negatives, namely false and missed alarms. L448 – 449 will be rephrased for clarity:

"For the representation of the results, the classification scheme typically adopted in the literature does not account for misclassification costs (Cantarino et al., 2019), meaning that the costs associated with false and missed alarms are implicitly assumed equal."

L459: This sentence about costs should come much earlier.

You are right, this theme has never been properly described within the manuscript. An introduction on the theme of misclassification costs will be added in the Introduction paragraph (please refer to comment on L94).

**Figures:**

Fig. 1: Coordinate grid is missing

Coordinate bar for Northern Italy was added.

Fig. 2. The last two classes in the cum. rainfall scale are incorrectly labeled.

Fig. 2 was changed to show the spatial distribution of maximum daily rainfall intensity, and the legend was changed accordingly.

Fig 4.: What are the numbers in parentheses? For lithology, there is the yellow bar for the 2014-2019 comparison missing.

Number in parentheses refer to the total number of variables that was included in each group. The explanation of this detail was added into the caption. The yellow bar for lithology is not actually missing, but it represents a 0% because between 2014 and 2019 there are no dissimilarities between the variables distribution.

Fig.5: see comment above. Why are there four correlation coefficients given but there are only 3 different data sets, i.e. the last (black) one is confusing.

The black coefficient was referred to the landslide density across all the inventories, but has been removed as suggested.

Fig. 8: The scale bar given here seems to have the same length wrt the maps as the one shown in Fig. 2, yet here it's 24 km whereas in Fig. 2 it is 32 km long.

The scale of the two maps is different (Fig. 2 → 1:800000, Fig..8 → 1:600000), thus leading to scale bars with the same length but different values.

Table 1. Reclassification scheme of the original lithological units from the geological map of Piemonte Region, at scale 1:250,000 (Piana et al., 2017).

| Litho-stratigraphical unit | Lithological description | Reclassified lithology |
|---|---|---|
| Bacino di Alessandria | Fluvial deposits | Gravels and Sands |
| Villafranchiano c: Unita di Maranzana | Sands, Gravels, Clays | Sands and Gravels |
| Formazione di Cassano Spinola | Sandstones, Conglomerates | Sandstones and Conglomerates |
| Sabbie di Asti b | Sands, Gravels | Sands and Gravels |
| Marne di S. Agata Fossili | Carbonate rich mudstones, Siltstones | Marls |
| Formazione di Serravalle | Sandstones, Siltstones | Sandstones and Siltstones |
| Villafranchiano b: Sabbie di Ferrere e Silt di S.Martino | Sands, Gravels, Siltstones | Sands and Gravels |
| Complesso caotico della Valle Versa | Gypsums or Anhydrites, Limestones, Clays | Gypsum |
| Marne di Cessole | Carbonate rich mudstones, Sandstones | Marls |
| Complesso caotico di Rocca Grimalda | Sandstones | Sandstone Breccias |
| Formazione di Visone | Sandstones, Impure limestones | Sandstones and Marls |
| Formazione di Molare | Conglomerates, Sandstones | Sandstones and Conglomerates |
| Formazione di Rigoroso | Carbonate rich Mudstones, Sandstones | Marls |
| Formazione di Cortemilia, Formazione di Costa Areasa | Sandstones, Mudstones | Sandstones and Siltstones |
| Serpentiniscisti antigoritici del Bric del Dente | Serpentinites | Serpentinites |
| Membro delle arenarie di Cassinelle, Formazione di Rigoroso | Sandstones, Conglomerates | Sandstones and Conglomerates |
| Formazione di Bistagno | Carbonate rich mudstones, Sandstones | Sandstones and Marls |
| Membri di Rocca Crovaglia, di Ronchi e di C. Garino, Formazione di Costa Montada | Arenites, Carbonate rich mudstones | Sandstones and Marls |
| Membro di Cascina Colombara, Formazione di Costa Montada | Carbonate rich mudstones | Marls |
| Calcari di Voltaggio | Limestones | Limestones |
| Metabasiti di Rossiglione | Chlorite actinolite epidote metamorphic rocks, Amphibolites, Schists | Prasinites |
| Calcescisti del Turchino | Schists, chlorite actinolite epidote metamorphic rocks | Calcschists |
| Calcari di Gallaneto | Limestones | Limestones |

| | | |
|---|---|---|
| Peridotiti lherzolitiche del Monte Tobbio | Peridotites | Gabbros and Peridotites |
| Metagabbri eclogitici della Colma, Metagabbri del Bric Mazzapiede, Metagabbri eclogitici di Prato del Gatto | Gabbros, Eclogites | Gabbros and Peridotites |
| Scisti filladici del Monte Larvego | Slates, Limestones | Slates |
| Dolomie del M. Gazzo | Dolomites | Limestones |
| Metabasalti di Cravasco | Basalts | Basalts |
| Serpentiniti di Case Bardane | Serpentinites | Serpentinites |
| Metagabbri del Monte Lecco | Gabbros | Gabbros and Peridotites |
| Argilloscisti di Costagiutta | Slates, Limestones | Slates |
| Serpentiniti del Bric dei Corvi | Serpentinites | Serpentinites |
| Argilloscisti di Murta | Slates | Slates |
| Metabasiti - Unita Figogna | Basalts, Breccias | Basalts |
| Argille azzurre a | Carbonate rich Mudstones, Silts, Sands, Gravels | Marls and Sands |
| Argille Azzurre b | Carbonate rich Mudstones, Silts, Sands | Marls and Sands |

**Bibliography**

Alvioli, M., Marchesini, I., Reichenbach, P., Rossi, M., Ardizzone, F., Fiorucci, F., and Guzzetti, F.: Automatic delineation of geomorphological slope units with r.slopeunits v1.0 and their optimization for landslide susceptibility modeling, Geosci. Model Dev., 9, 3975–3991, https://doi.org/10.5194/gmd-9-3975-2016, 2016.

ARPA Piemonte and Regione Piemonte: Analisi del clima regionale del periodo 1981-2010 e tendenze negli ultimi 60 anni, 2020.

Bogaard, T. and Greco, R.: Invited perspectives: Hydrological perspectives on precipitation intensity-duration thresholds for landslide initiation: Proposing hydro-meteorological thresholds, Nat. Hazards Earth Syst. Sci., 18, 31–39, https://doi.org/10.5194/nhess-18-31-2018, 2018.

Cantarino, I., Carrion, M. A., Goerlich, F., and Martinez Ibañez, V.: A ROC analysis-based classification method for landslide susceptibility maps, Landslides, 16, 265–282, https://doi.org/10.1007/s10346-018-1063-4, 2019.

Carrara, A., Cardinali, M., Detti, R., Guzzetti, F., Pasqui, V., and Reichenbach, P.: GIS techniques and statistical models in evaluating landslide hazard, Earth Surf. Process. Landforms, 16, 427–445, https://doi.org/10.1002/esp.3290160505, 1991.

Carrara, A., Crosta, G., and Frattini, P.: Comparing models of debris-flow susceptibility in the alpine environment, Geomorphology, 94, 353–378, https://doi.org/10.1016/j.geomorph.2006.10.033, 2008.

Gilbert, G. K.: Finley's Tornado Predictions, Am. Meteorol. J., 1, 166–172, 1884.

Guzzetti, F., Carrara, A., Cardinali, M., and Reichenbach, P.: Landslide hazard evaluation: a review of current techniques and their application in a multi-scale study, Central Italy, Geomorphology, 13, 1995, 1999.

Guzzetti, F., Peruccacci, S., Rossi, M., and Stark, C. P.: Rainfall thresholds for the initiation of landslides in central and southern Europe, Meteorol. Atmos. Phys., 98, 239–267, https://doi.org/10.1007/s00703-007-0262-7, 2007.

Heidke, P.: Berechnung Des Erfolges Und Der Güte Der Windstärkevorhersagen Im Sturmwarnungsdienst, Geogr. Ann., 8, 301–349, https://doi.org/10.1080/20014422.1926.11881138, 1926.

Hwang, C.-L. and Yoon, K.: Multiple Attribute Decision Making: methods and applications a state-of-the-art survey, Springer Sci. Bus. Media, 186, 2012.

Jacobs, L., Kervyn, M., Reichenbach, P., Rossi, M., Marchesini, I., Alvioli, M., and Dewitte, O.: Regional susceptibility assessments with heterogeneous landslide information: Slope unit- vs. pixel-based approach, Geomorphology, 356, 107084, https://doi.org/10.1016/j.geomorph.2020.107084, 2020.

Luino, F.: The Flood and Landslide Event of November 4-6 1994 in Piedmont Region ( Northwestern Italy ): Causes and Related Effects in Tanaro, 24, 123–129, 1999.

Mason, P. J. and Rosenbaum, M. S.: Geohazard mapping for predicting landslides: An example from the Langhe Hills in Piemonte, NW Italy, Q. J. Eng. Geol. Hydrogeol., 35, 317–326, https://doi.org/10.1144/1470-9236/00047, 2002.

Peirce, C. S.: The numerical measure of the success of predictions, Science, 453–454, 1884.

Smith, H. G., Neverman, A. J., Betts, H., and Spiekermann, R.: The influence of spatial patterns in rainfall on shallow landslides, Geomorphology, 437, 108795, https://doi.org/10.1016/j.geomorph.2023.108795, 2023.

Steger, S., Brenning, A., Bell, R., and Glade, T.: The propagation of inventory-based positional

errors into statistical landslide susceptibility models, Nat. Hazards Earth Syst. Sci., 16, 2729–2745, https://doi.org/10.5194/nhess-16-2729-2016, 2016.

Thomas, M. A., Mirus, B. B., and Collins, B. D.: Identifying Physics-Based Thresholds for Rainfall-Induced Landsliding, Geophys. Res. Lett., 45, 9651–9661, https://doi.org/10.1029/2018GL079662, 2018.

Tiranti, D., Nicolò, G., and Gaeta, A. R.: Shallow landslides predisposing and triggering factors in developing a regional early warning system, Landslides, 16, 235–251, https://doi.org/10.1007/s10346-018-1096-8, 2019.

Yule, G. U.: On the association of attributes in statistics, Philos. Trans. R. Soc. London. Ser. A, Contain. Pap. a Math. or Phys. Character, 194, 257–319, https://doi.org/10.1098/rsta.1900.0019, 1900.

---

## Author Comment (AC2)

**Response to Review 2 of "Shaping shallow landslide susceptibility as a function of rainfall events" by Fumagalli et al.**

We would like to thank the editor and the two reviewers for the detailed and valuable comments on this manuscript. We revised the entire draft and, based on the given suggestions, made changes within some paragraphs, to the indicated figures, and to the cross-references.

Below we address the specific comments and questions made by the 2 reviewer (for lines numbering please refer to the original manuscript).

Looking forward to your replies,

Micol Fumagalli, on behalf of all co-authors
* * *
Review of the manuscript #NHESS-2024-140 entitled "Shaping shallow landslide susceptibility as a function of rainfall events" by Micol Fumagalli and colleagues.

The manuscript entitled "Shaping shallow landslide susceptibility as a function of rainfall events" by Micol Fumagalli et al. presents an interesting rainfall-based shallow landslide susceptibility analysis, in Orba basin at Piedmont Region, Italy, using logistic regression. Five extreme rainfall events (3 with landslides, and 2 without) have been considered for both training and validation purposes. Three susceptibility scenarios have been performed for each of the five rainfall events.

This work consists of a good exercise for slope instability analysis, considering the modelling and validation processes, being also useful for civil protection warning and mitigation purposes.

This is an original and very good work. Nevertheless, there are few minor issues that should be considered.

Thus, I suggest accept with minor revisions.

Next I put some issues:

L. 158-159: Please, revise and correct the legend of cumulative rainfall, in Fig. 2.

Also following the suggestions of the other reviewer, rainfall maps in Figure 2 will be changed to show the maximum daily rainfall intensity,

L. 167: Frattini and Crosta (2013), instead of "(Frattini and Crosta, 2013).

The reference in L167 will be corrected as suggested.

L. 196-201: Which was the interpolation method and which is the resolution of the interpolated raster retrieved from rainfall data?

We used a Natural Neighbour interpolator (Sibson, 1981), which preserves the original values at the sample points and is less affected by biases that could be induced when data form spatial clusters. The final maps were produced at a resolution of 5 m, to match the resolution of the topographical information.

L197 will be modified as follows:

"These parameters were obtained by interpolating daily rainfall data collected at 39 and 51 gauging stations for the 1977 and 2014/2019 rainfall events, respectively, with a natural neighbour technique, at a spatial resolution of 5 m."

L. 203: when you refer to study areas, do you mean the areas where the meteorological stations are, or do you refer to the different lithological subdivisions?

In L203 "areas" was a typo. In this line we refer to the Orba basin, meaning that the normalization values were calculated for each pixel within the basin. L203 will be changed accordingly.

L. 358: Please explain the meaning of "Ci".

In L358, "Ci" represents the relative closeness degree of each alternative to the ideal solution, but since it is unnecessary to the understanding of Figure 7, it will be removed from the caption.

L. 388: Could you write in parentheses the meaning of VL (very low), L (low), M (...), H, VH in Table 2, like you did for HST?

The meaning of the abbreviations will be added in the caption of Table 2, as suggested.